# FractalFold: Towards Fractal Structure Modeling for Hierarchical Inverse Protein Folding

## Abstract

Inverse protein folding aims to design amino acid sequences that fold into desired backbone structures, representing a long-standing challenge in computational protein design. While recent deep learning approaches have achieved significant progress, existing methods predominantly treat protein structures as flattened sequences, overlooking their inherent hierarchical and fractal organization. To address this limitation, we propose FractalFold, a novel transformer-based model that performs structure-informed inverse folding by recursively invoking multi-level atomic fractal transformers. FractalFold employs a coarse-to-fine sequence refinement paradigm that mirrors the intrinsic hierarchical nature of protein structures. To generalize our approach to quasi-fractal proteins with variable-length structural segments, we introduce the Hierarchical Fractal Segmentation Module (HFSM), which leverages attention patterns from pre-trained protein language models to recursively partition protein structures into tree-organized patches. Extensive experiments on the CATH benchmarks demonstrate that FractalFold achieves state-of-the-art performance in sequence recovery rate and perplexity while generating sequences with enhanced foldability, establishing a new paradigm for structure-informed protein design.

## 1 Introduction

Proteins are amino acid sequences that control biological processes such as transcription, translation, and immune response. Designing novel proteins that fold into desired 3D structures, also termed protein inverse folding, remains one of the fundamental challenges in computational biology with important applications in protein engineering, drug design, and synthetic biology.

Beyond traditional physics-based methods like Rosetta, which suffer from high computational costs, deep learning-based methods have emerged as promising alternatives. Despite substantial advancement, a major challenge in inverse protein folding remains the extraction of hierarchical structural patterns from raw 3D coordinate sequences and their explicit incorporation as inductive biases into the model architecture. As shown in Figure 1, transformer-based models Ren et al. (2024b) directly encode the coordinate sequence as a flattened token sequence and perform self-attention calculations autoregressively, which fails to capture the inherent semantics of protein structures and inevitably results in error accumulation. Recent methods harness graph-based structure encoders alongside diffusion models Yi et al. (2023a), GNNs Gao et al. (2023), and transformers Zheng et al. (2023a); Ingraham et al. (2019a). Nonetheless, they only model the lowest-level structural bias by aggregating information from nearest residues using kNN algorithms Fix (1985) during encoding, still lacking a way to explicitly incorporate the prior of the full hierarchical protein structure.

To address the above issue, we propose to predict amino acid sequences by recursively invoking multi-level atomic fractal transformers, which naturally align with protein's inherent hierarchical organization through the model architecture. Our proposal is analogous to the concept of **fractals** in mathematics, which was first employed by FractalGen Li et al. (2025) in image generation tasks. Compared to images, protein structures display stronger self-similarity and recursive organizational patterns across multiple scales—from secondary structure motifs to functional domains and complete tertiary folds Stapleton et al. (1980); Ikeda et al. (1999); Sendker et al. (2024). As reported in

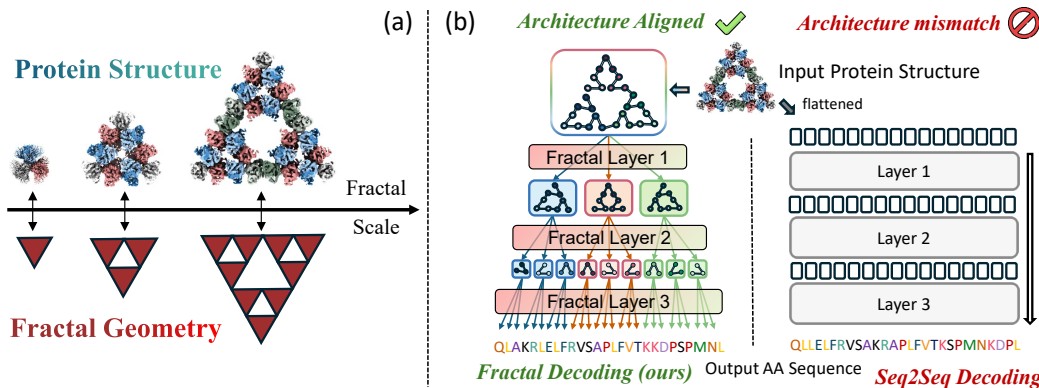

Figure 1: (a) Protein structures exhibit fractal geometry Sendker et al. (2024) with self-similar patterns across multiple scales. (b) The architecture of FractalFold is aligned with protein structure (left), while conventional methods lose structural prior by flattening 3D structures into sequential tokens (right).

Enright & Leitner (2005), 200 proteins from the Protein Data Bank Berman et al. (2000) ranging from 100 to over 10,000 amino acids have an average fractal dimension of 2.5. Building on the fractal property of protein structure, we propose **FractalFold**, a fractal transformer model specifically tailored for hierarchical-structure-informed inverse protein folding, which inherently conforms to the fractal nature of proteins. FractalFold adopts a coarse-to-fine refinement paradigm that effectively reduces error accumulation with one-shot inference.

To further generalize FractalFold to broader quasi-fractal proteins containing recursive units of varying lengths, we introduce the **Hierarchical Fractal Segmentation Module (HFSM)**, a structure-aware dynamic decomposition algorithm that exploits attention patterns from pre-trained protein language models (pLMs) to recursively segment protein sequences into variable-length patches. Extensive evaluation on CATH benchmarks demonstrates that FractalFold achieves state-of-the-art performance across comprehensive metrics, including sequence recovery rate, perplexity and structural foldability. By establishing a fractal-based coarse-to-fine refinement framework for protein design, our work opens a new paradigm for structure-informed inverse folding that aligns with the intrinsic hierarchical organization of biological proteins.

The primary contributions of this work are threefold:

1. We introduce FractalFold, a fractal inverse protein folding model that explicitly captures the hierarchical structural bias of proteins by recursively invoking atomic transformer units.

2. We develop HFSM, a novel dynamic segmentation algorithm that enables adaptive multi-scale structure decomposition, generalizing FractalFold to broader quasi-fractal proteins with variable-length recursive units.

3. We achieve state-of-the-art experimental performance on established benchmarks with significant improvements in sequence recovery, perplexity, and foldability compared to existing methods.

## 2 PRELIMINARIES

### 2.1 PROBLEM FORMULATION

Let $\mathbf{S} \in \mathbb{R}^{N \times N_{atom} \times 3}$ denote the 3D coordinates of backbone atoms for a protein of length $N$, where $\mathbf{S}_i \in \mathbb{R}^{N_{atom} \times 3}$ represents the atomic coordinates of the $i$-th residue. Let $\mathbf{A} = (a_1, a_2, \ldots, a_N)$ denote the corresponding amino acid sequence, where each $a_i \in \{1, 2, \ldots, 20\}$ is an integer index representing one of the 20 standard amino acids.

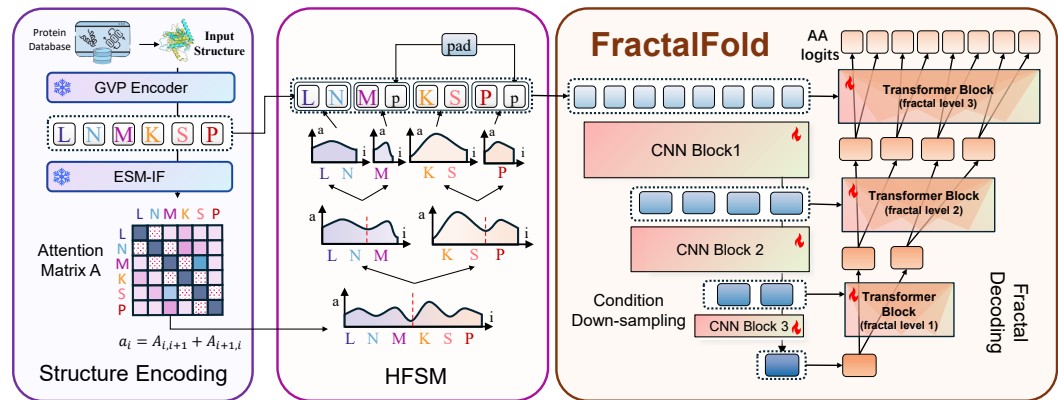

Figure 2: Overview of FractalFold architecture. The input protein structure is encoded using GVP and ESM-IF to extract geometric and contextual features. The Hierarchical Fractal Segmentation Module (HFSM) decomposes the sequence into multi-scale segments using attention-based break-point selection. The Fractal Transformer processes these hierarchical representations through cascaded blocks, with each fractal level conditioning on the previous scale to generate amino acid sequences via coarse-to-fine prediction.

The neural network $f_\theta$ performs the mapping:

$$f_\theta : \mathbb{R}^{N \times N_{\text{atom}} \times 3} \to \Delta^{20 \times N}, \tag{1}$$

where $\Delta^{20 \times N}$ represents the space of probability distributions over amino acid sequences of length $N$.

The model parameters $\theta$ are optimized to maximize the conditional log-likelihood:

$$\mathcal{L}(\theta) = \sum_{(\mathbf{S}, \mathbf{A}) \in \mathcal{D}} \log p(\mathbf{A} \mid \mathbf{S}; \theta), \tag{2}$$

where $\mathcal{D}$ denotes the training dataset consisting of structure-sequence pairs.

## 2.2 HIERARCHICAL PROTEIN MODELING

FractalFold decomposes the sequence generation task through recursive segmentation. Given a protein structure $S$ of length $N$, we construct a $K$-scale hierarchy where each scale partitions the sequence into segments of manageable size. At scale $k \in \{1, 2, \ldots, K\}$, we divide the sequence into $L_k$ segments.

Let $T^{(k)} = \{\mathbf{t}_1^{(k)}, \mathbf{t}_2^{(k)}, \ldots, \mathbf{t}_{L_k}^{(k)}\}$ denote the $L_k$ padded segments at scale $k$, where each segment $\mathbf{t}_m^{(k)} = (s_{b_m}, s_{b_{m+1}}, \ldots, s_{e_m})$ contains a structure subsequence spanning from position $b_m$ to $e_m$. Similarly, we define $H^{(k)} = \{\mathbf{h}_1^{(k)}, \mathbf{h}_2^{(k)}, \ldots, \mathbf{h}_{L_k}^{(k)}\}$ as the corresponding structural features for segments in $T^{(k)}$.

The hierarchical generation follows a coarse-to-fine strategy:

$$p(A \mid S) = p(\mathcal{T}^{(1)} \mid \mathcal{H}^{(1)}) \prod_{k=1}^{K-1} p(\mathcal{T}^{(k+1)} \mid \mathcal{H}^{(k+1)}, \mathcal{T}^{(k)}) \quad . \tag{3}$$

This decomposition enables progressive refinement from global structural patterns to local amino acid discrimination.

# 3 METHOD

## 3.1 FRAMEWORK OVERVIEW

FractalFold employs a hierarchical generation strategy that decomposes protein inverse folding into multiple scales of conditional prediction. As illustrated in Figure X, the input protein backbone structure $S$ is first processed by a GVP-based encoder Hsu et al. (2022a) to capture geometric features with spatial neighborhood information. The Hierarchical Fractal Segmentation Module (HFSM) then organizes both structural representations and target amino acid sequences into $K$ hierarchical scales: $\{\mathcal{H}^{(k)}, \mathcal{T}^{(k)}\}_{k=1}^{K}$, where each scale represents segments of decreasing granularity.

The fractal generator performs coarse-to-fine prediction through cascaded conditional generation. Starting from the coarsest scale, each scale $k$ generates $\mathcal{T}^{(k)}$ by conditioning on the corresponding structural features $\mathcal{H}^{(k)}$ and previously generated sequence representations $\mathcal{T}^{(k-1)}$. The final AA sequence is obtained by sampling from the predicted logits at the finest scale, ensuring that global structural constraints guide local amino acid selection while maintaining consistency across scales.

## 3.2 HIERARCHICAL FRACTAL SEGMENTATION MODULE (HFSM)

The HFSM transforms the linear protein sequence into a hierarchical tree-like structure by identifying optimal breakpoints that maximize intra-segment coherence. Given the input structural sequence, we first obtain contextual representations using a pretrained ESM2 protein language model and obtain the attention matrix $\mathbf{A} \in \mathbb{R}^{N \times N}$. We then compute link probabilities between adjacent residues using neighbor attention:

$$a_i = \sqrt{\mathbf{A}_{i,i+1} \times \mathbf{A}_{i+1,i}} \quad , \tag{4}$$

where $a_i$ represents the bidirectional link probability that residues $i$ and $i + 1$ belong to the same structural unit.

---

**Algorithm 1** Hierarchical Fractal Segmentation

**Require:** Sequence $S = (s_1, s_2, \ldots, s_N)$, ESM2 model, fractal scales $K$, segment counts $\{L_k\}_{k=1}^{K}$
**Ensure:** Hierarchical segmentation $\{T^{(k)}\}_{k=1}^{K}$
1: Compute attention matrix $\mathbf{A} = ESM2(S)$
2: Compute link probabilities: $a_i = \sqrt{\mathbf{A}_{i,i+1} \times \mathbf{A}_{i+1,i}}$ for $i = 1, \ldots, N-1$
3: Initialize $T^{(0)} = \{(1, N)\}$ {Full sequence as single segment}
4: **for** $k = 1$ to $K$ **do**
5:    $r = L_k/L_{k-1} - 1$ {New breakpoints per segment}
6:    $T^{(k)} = \emptyset$
7:    **for** each segment $(b_m, e_m) \in T^{(k-1)}$ **do**
8:       Find $r$ positions with minimum $a_i$ in $[b_m, e_m - 1]$
9:       Split segment $(b_m, e_m)$ at these $r$ positions
10:      Add resulting sub-segments to $T^{(k)}$
11:    **end for**
12: **end for**
13: Pad all segments in $T^{(K)}$ to uniform length with $\langle \text{PAD} \rangle$
14: **return** $T^{(K)}$

---

For a segment spanning positions $[b_m, e_m]$, the segment coherence is defined as the product of all link probabilities within the segment: $\prod_{i=b_m}^{e_m-1} a_i$. To find the optimal segmentation at scale $k$ with $L_k$ segments, we solve:

$$T^{(k)*} = \arg\min_{T^{(k)}} \sum_{m=1}^{L_k} \sum_{i=b_m}^{e_m-1} (-\log a_i) \quad . \tag{5}$$

where $T^{(k)} = \{\mathbf{t}_1^{(k)}, \mathbf{t}_2^{(k)}, \ldots, \mathbf{t}_{L_k}^{(k)}\}$ represents the set of segments at scale $k$. We prove in the Appendix G that this optimization is equivalent to selecting the $L_k - 1$ positions with the lowest link

probabilities as breakpoints. $T^{(0)}$ is defined as the structural representation obtained by pre-trained structure encoder. The algorithm iteratively constructs the tree structure by selecting breakpoints within existing segments from the previous scale. We first employ the Geometric Vector Perceptron (GVP) encoder Hsu et al. (2022a) to process the input structure sequence. The GVP encoder adopt the kNN algorithm to fuse neighboring information around each residue while maintaining rotation invariance and equivariance properties. Starting from the full sequence at scale 0, at each scale $k$, we identify the $L_k - L_{k-1}$ positions with minimum link probabilities within the current segments and introduce new breakpoints accordingly. To maintain balanced tree growth, we add an equal number of breakpoints to each segment from the previous scale: specifically, $L_k/L_{k-1} - 1$ new breakpoints per segment. The process continues until reaching the top fractal scale $K$, creating a hierarchical structure $\{T^{(k)}\}_{k=1}^{K}$ where each scale contains segments of progressively finer granularity. Finally, segments at fractal scale $K$ are padded to uniform length with special token $\langle \text{PAD} \rangle$ to enable efficient batch processing in the subsequent fractal transformer, resulting in padded segments $\mathbf{t}_m^{(K)} = (s_{b_m}, s_{b_m+1}, \ldots, s_{e_m}, \langle \text{PAD} \rangle, \ldots, \langle \text{PAD} \rangle)$. The complete hierarchical segmentation algorithm is presented in Algorithm 1.

---

**Algorithm 2** FractalFold Training Algorithm

---

**Require:** Protein dataset $\mathcal{D}$, fractal scales $K$, FractalFold model parameters $\theta$
**Ensure:** Trained model parameters $\theta$
1: **for** each batch $(S, A) \in \mathcal{D}$ **do**
2:     $T^{(1)}, \ldots, T^{(K)} \leftarrow \text{HFSM}(S, K)$                            {Hierarchical segmentation}
3:     $\mathbf{c}^{(0)} \leftarrow \mathbf{0}$                                               {Initialize conditioning}
4:     **for** $k = 1$ to $K$ **do**
5:        $\mathbf{S}^{(k)} \leftarrow \text{Compress}(T^{(k)})$                        {Multi-scale compression}
6:        $\mathbf{X}^{(k)} \leftarrow [\mathbf{S}^{(k)}, \mathbf{c}^{(k-1)}]$                            {Add conditioning}
7:        $\mathbf{Y}^{(k)} \leftarrow \text{Transformer}^{(k)}(\mathbf{X}^{(k)})$               {Fractal transformer}
8:        $\mathbf{c}^{(k)} \leftarrow \text{Reshape}(\mathbf{Y}_{1:L_k}^{(k)})$                 {Update conditioning}
9:     **end for**
10:    $\mathcal{L} \leftarrow -\sum_{i=1}^{L_K} \log P(y_i | \mathbf{Y}_i^{(K)})$             {Cross-entropy loss}
11:    $\theta \leftarrow \text{Update}(\theta, \nabla_\theta \mathcal{L})$                       {Backprop & update}
12: **end for**
13: **return** $\theta$

---

## 3.3 FRACTAL TRANSFORMER

After obtaining the hierarchically padded segmentation $T^{(K)}$ from HFSM, a multi-scale compressor module transforms it into $K$ fractal scales with shapes $[B, L_1, D]$, $[B, L_2, D]$, ..., $[B, L_K, D]$, where $B$ is the batch size, $L_k$ represents the sequence length at fractal scale $k$, and $D$ is the feature dimension.

The fractal transformer processes these multi-scale representations in a hierarchical manner. At each fractal scale $k$, the transformer takes as input the $L_k$ structural tokens along with one conditioning token from the previous fractal scale $k - 1$. This conditioning mechanism enables information flow across different granularity scales, allowing coarse-scale structural patterns to guide fine-scale predictions.

Specifically, for fractal scale $k$, the input consists of:

$$\mathbf{X}^{(k)} = [\mathbf{s}_1^{(k)}, \mathbf{s}_2^{(k)}, \ldots, \mathbf{s}_{L_k}^{(k)}, \mathbf{c}^{(k-1)}] \quad , \tag{6}$$

where $\mathbf{s}_i^{(k)}$ represents the $i$-th structural token at scale $k$, and $\mathbf{c}^{(k-1)}$ is the conditioning token from scale $k - 1$. The transformer processes this input through standard self-attention and feed-forward layers, producing output predictions:

$$\mathbf{Y}^{(k)} = \text{Transformer}^{(k)}(\mathbf{X}^{(k)}) \in \mathbb{R}^{B \times (L_k+1) \times D} \quad , \tag{7}$$

After dropping the last token, $\mathbf{Y}^{(k)}$ is reshaped to form the conditioning token for the next fractal scale with shape $[B \cdot L_k, 1, D]$. This iterative process continues through all fractal scales, with each scale providing hierarchical guidance to the next.

The output of the final fractal scale $K$ is used to compute the cross-entropy loss:

$$\mathcal{L} = -\sum_{i=1}^{L_K} \log P(y_i|\mathbf{Y}_i^{(K)}) \quad . \tag{8}$$

where $y_i$ is the ground truth token at position $i$, and $\mathbf{Y}_i^{(K)}$ is the corresponding predicted distribution from the transformer output at the finest scale.

This hierarchical processing strategy captures structural patterns at multiple scales while maintaining computational efficiency through fractal decomposition. By adopting this "divide-and-conquer" approach, the computational complexity is effectively reduced from $O(N^2)$ to $O(NlogN)$, which is proved in Appendix H. The complete training algorithm is presented in Algorithm 2.

## 4 EXPERIMENTS

In this section, we evaluate FractalFold against 10 state-of-the-art baselines on standard protein inverse folding benchmarks, analyze refoldability to assess sequence-structure consistency, and conduct ablation studies to validate our key design choices.

### 4.1 DATASETS AND BASELINES

To fully evaluate FractalFold's performance, we conduct extensive experiments comparing FractalFold to 10 baseline methods, which can be organized into three distinct classes depending on the decoding mode. **Autoregressive models** include StructGNN (Ingraham et al., 2019b), Graph-Trans (Ingraham et al., 2019b), GCA (Tan et al., 2023), GVP (Jing et al., 2021), AlphaDesign (Gao et al., 2022), ESM-IF (Hsu et al., 2022b), and ProteinMPNN (Dauparas et al., 2022). **One-shot model**: PiFold (Gao et al., 2023) **Diffusion-based model** includes GraDe-IF (Yi et al., 2023b), which employ probabilistic denoising processes for sequence generation.

The evaluation is conducted on the widely-used **CATH v4.2** and **CATH v4.3** protein structure datasets. For consistency and fair comparison, we adopt the standard data partitions previously established in the literature. The CATH v4.2 split (Ingraham et al., 2019b) provides 18,024 structures for training, 608 for validation, and 1,120 for testing. Similarly, the CATH v4.3 split (Hsu et al., 2022b) allocates 16,153 structures for training, 1,457 for validation, and 1,797 for testing.

### 4.2 EVALUATION METRICS

Model performance was quantified using **perplexity** and amino acid **recovery rate**, which are standard metrics for this task (Hsu et al., 2022b). To provide a granular analysis, we report the median recovery rate and perplexity across three subsets of the test data: all proteins, single-chain proteins only, and short proteins (defined as having a length $\leq 100$ residues).

### 4.3 TRAINING DETAILS

All experiments were conducted on four NVIDIA GTX 4090 GPUs. Models were trained for 50 epochs using AdamW optimizer with a base learning rate of $5 \times 10^{-5}$. The architecture employs a frozen GVP structural encoder with multi-resolution encoding at three scales and fractal hierarchy using progressive patch refinement from coarse to fine levels. More training details are provided in Appendix F.

### 4.4 MAIN RESULTS ON INVERSE FOLDING

We evaluate FractalFold against state-of-the-art methods on the CATH 4.2 and 4.3 benchmarks. Table 1 demonstrates that FractalFold achieves superior performance across all metrics and protein categories. On CATH 4.2, FractalFold attains 53.62% recovery rate and 4.27 perplexity, outperforming the previous best method GraDe-IF (52.21% recovery, 4.35 perplexity). On CATH 4.3, our method achieves 52.23% recovery rate and 3.81 perplexity, surpassing even ESM-IF augmented with 1.2M AlphaFold2 structures (51.60% recovery, 4.01 perplexity) while using only standard training data.

Table 1: Experimental Results of FractalFold and 10 baselines on the CATH Benchmark. Partial baseline results are quoted from Hsu et al. (2022b); Yi et al. (2023b). †: "Single-chain" in Hsu et al. (2022b) is defined differently. The **best** results are bolded and suboptimal results are underlined.

| | Model | Perplexity ↓ | | | Recovery Rate % ↑ | | |
|---|---|---|---|---|---|---|---|
| | | Short | Single-chain | All | Short | Single-chain | All |
| CATH 4.2 | StructGNN (Ingraham et al., 2019b) | 8.29 | 8.74 | 6.40 | 29.44 | 28.26 | 35.91 |
| | GraphTrans (Ingraham et al., 2019b) | 8.39 | 8.83 | 6.63 | 28.14 | 28.46 | 35.82 |
| | GCA (Tan et al., 2023) | 7.09 | 7.49 | 6.05 | 32.62 | 31.10 | 37.64 |
| | GVP (Jing et al., 2021) | 7.23 | 7.84 | 5.36 | 30.60 | 28.95 | 39.47 |
| | AlphaDesign (Gao et al., 2022) | 7.32 | 7.63 | 6.30 | 34.16 | 32.66 | 41.31 |
| | ProteinMPNN (Dauparas et al., 2022) | 6.21 | 6.68 | 4.61 | 36.35 | 34.43 | 45.96 |
| | PiFold (Gao et al., 2023) | 6.04 | 6.31 | 4.55 | 39.84 | 38.53 | 51.66 |
| | GraDe-IF (Yi et al., 2023b) | 5.49 | 6.21 | 4.35 | 45.27 | 42.77 | 52.21 |
| | **FractalFold** (ours) | **5.25** | **6.08** | **4.27** | **46.19** | **44.28** | **53.62** |
| CATH 4.3 | GVP-large (Hsu et al., 2022b) | 7.68 | 6.12† | 6.17 | 32.60 | 39.40† | 39.20 |
| | ESM-IF (Hsu et al., 2022b) | 8.18 | 6.33† | 6.44 | 31.30 | 38.50† | 38.30 |
| | +1.2M AF2 predicted data | 6.05 | 4.00† | 4.01 | 38.10 | 51.50† | 51.60 |
| | **FractalFold** (ours) | **5.45** | **4.57** | **3.81** | **42.92** | **52.47** | **52.23** |

The performance gains are most pronounced for short proteins ($\leqslant 100$ residues), suggesting that hierarchical fractal decomposition is particularly effective for compact structures. The consistent improvements across both metrics indicate that our coarse-to-fine refinement paradigm successfully reduces error accumulation inherent in autoregressive approaches.

The superior performance stems from FractalFold's multi-scale one-shot decoding strategy, which addresses the error accumulation problem in autoregressive methods including StructGNN (Ingraham et al., 2019b), GraphTrans (Ingraham et al., 2019b), GCA (Tan et al., 2023), and GVP (Jing et al., 2020). By simultaneously generating amino acid tokens within a "divide-and-conquer" fractal decoding framework, FractalFold effectively prevents error propagation across different structural domains, achieving consistently lower perplexity scores across all protein categories.

This result is particularly significant as it demonstrates that architectural innovation can outperform data scaling. While ESM-IF requires massive synthetic data augmentation from AlphaFold2 predictions, FractalFold achieves superior performance through its hierarchical fractal modeling alone. This suggests that capturing the intrinsic structural organization of proteins is more effective than brute-force data expansion.

## 4.5 REFOLDABILITY ANALYSIS

Refoldability measures whether generated sequences can fold back to structures similar to the target backbone, serving as a critical evaluation of sequence-structure consistency in protein design. Following the evaluation configuration in Wang et al. (2023), we use a high-quality test set of 82 samples to assess the foldability of predicted sequences. The toolkit from Zhang & Tm-Align is used to calculate Ref-TM and Ref-pLDDT scores for protein structures folded by ESMFold Lin et al. (2022), OmegaFold Wu et al. (2022), and AlphaFold2 Jumper et al. (2021). We compare FractalFold against seven baselines, including pLM-adapted methods (ByProt Zheng et al. (2023b), AF-Design Wang et al. (2022), and ESM-Design Verkuil et al. (2022)).

The results are summarized in Table 2. FractalFold achieves superior refoldability performance, attaining the highest TM scores of 0.81 (ESMFold) and 0.92 (AlphaFold2), along with the best pLDDT scores of 77.81 (ESMFold) and 88.24 (AlphaFold2). Notably, FractalFold surpasses ProteinMPNN, the previous leader in refoldability metrics, while simultaneously achieving the highest recovery rate of 52.33%. These results demonstrate that FractalFold significantly outperforms pLM-based methods, highlighting the effectiveness of our structure-informed decoding mechanism in capturing essential sequence-structure relationships.

The results reveal several key insights. First, the exceptional structural fidelity demonstrates the effectiveness of our structure-informed decoding mechanism. FractalFold's generated sequences

Table 2: Refoldability results on the CATH dataset. Best and suboptimal results are **bolded** and underlined. We use TM and pLDDT to represent Ref-TM and Ref-pLDDT.

| Design method | ESMFold | | OmegaFold | | AlphaFold2 | | Recovery Rate% ↑ |
|---|---|---|---|---|---|---|---|
| | TM↑ | pLDDT↑ | TM↑ | pLDDT↑ | TM↑ | pLDDT↑ | |
| Uniform | 0.05 | 27.68 | 0.05 | 31.53 | 0.06 | 33.68 | 5.00 |
| Natural frequencies | 0.07 | 30.53 | 0.07 | 35.59 | 0.06 | 35.02 | 5.84 |
| *with pLM* | | | | | | | |
| ByProt (Zheng et al., 2023b) | 0.73 | 72.12 | 0.70 | 77.58 | 0.85 | 87.26 | 51.23 |
| AF-Design (Wang et al., 2022) | 0.53 | 61.37 | 0.53 | 72.04 | 0.52 | 75.29 | 15.95 |
| ESM-Design (Verkuil et al., 2022) | 0.38 | 59.65 | 0.38 | 62.66 | 0.37 | 60.02 | 17.33 |
| *without pLM* | | | | | | | |
| StructTrans (Ingraham et al., 2019b) | 0.72 | 68.85 | 0.64 | 70.35 | 0.79 | 80.66 | 35.89 |
| GVP Jing et al. (2020) | 0.73 | 69.67 | 0.67 | 74.33 | 0.83 | 84.29 | 39.46 |
| ProteinMPNN (Dauparas et al., 2022) | 0.80 | 76.53 | **0.76** | **80.75** | 0.87 | 87.89 | 41.44 |
| PiFold (Gao et al., 2023) | 0.71 | 67.55 | 0.64 | 70.21 | 0.82 | 82.54 | 44.86 |
| **FractalFold** (ours) | **0.81** | **77.81** | 0.74 | 79.23 | **0.92** | **88.24** | **52.33** |

consistently fold back to structures highly similar to the target backbone across all three prediction models, validating that our fractal decoding strategy successfully preserves critical structural information during sequence generation.

Second, FractalFold significantly surpasses methods utilizing protein language models (pLMs). This superiority highlights that our structure-centric hierarchical modeling captures essential sequence-structure relationships more effectively than pLM adapters.

Figure 3: FractalFold prediction for protein 1rtu.A (114 residues) showing high-confidence structural model with average pLDDT of 92.2. Structure views (left) and confidence-colored representations (right) demonstrate excellent prediction quality across most residues, with performance metrics including pTM score of 0.860 and FractalFold Score of 0.667. Notable confidence dip observed around residue 60 in the per-residue confidence plot.

## 4.6 CASE STUDY

We visualize the protein folded by the prediction results of FractalFold, as presented in Figure 3. Protein 1rtu.A serves as an exemplary case study demonstrating FractalFold's prediction capabilities on a 114-residue protein from the CATH 4.2 test set. The model achieved strong overall performance with an average pLDDT score of 92.2, pTM score of 0.860, and FractalFold score of 0.667, indicating high-quality structural prediction. The confidence-colored visualization reveals predominantly high-confidence regions (red coloring, pLDDT > 90) across most of the structure, with well-defined secondary structure elements clearly visible in both front and side views. Notably, the per-residue confidence plot identifies a localized confidence dip around residue 60 where

pLDDT drops to approximately 50, likely corresponding to a flexible loop region that presents inherent structural ambiguity. This confidence decrease is isolated and does not propagate to neighboring regions, as evidenced by the rapid recovery to high confidence levels, demonstrating FractalFold's ability to maintain overall structural integrity while appropriately flagging uncertain regions. The successful prediction of both terminal regions and the compact, well-organized three-dimensional fold topology validates the robustness of our approach across diverse structural elements within a single protein.

## 4.7 ABLATION STUDIES

To validate the effectiveness of key components in FractalFold, we conduct comprehensive ablation studies on the CATH v4.2 dataset, examining three critical design choices: pre-trained encoder initialization, one-shot decoding mechanism, and hierarchical fractal segmentation module (HFSM). The results are presented in Table 3.

Table 3: Ablation studies of key components on CATH v4.2. "w/ pre-training" uses pre-trained GVP encoder. "w/ one-shot" adopts one-shot decoding mechanism. "w/ HFSM" "applies the hierarchical segmentation module to the ground truth AA sequences.

| Encoder | Decoder | Segmentation | Perplexity ↓ | | | Recovery Rate % ↑ | | |
|---|---|---|---|---|---|---|---|---|
| w/ pre-training | w/ one-shot | w/ HFSM | Short | Single-chain | All | Short | Single-chain | All |
| | ✓ | ✓ | 5.48 | 6.27 | 4.31 | 44.67 | 44.10 | 52.10 |
| ✓ | | ✓ | 6.34 | 6.68 | 4.53 | 43.47 | 42.62 | 50.39 |
| ✓ | ✓ | | 6.77 | 6.43 | 4.34 | 42.35 | 43.57 | 51.24 |
| ✓ | ✓ | ✓ | **5.25** | **6.08** | **4.27** | **46.19** | **44.28** | **53.51** |

1. **Pre-trained Encoder:** Removing pre-trained GVP encoder initialization leads to consistent performance degradation across all protein categories. This demonstrates that pre-trained structural representations provide essential geometric priors that enable the model to better understand multi-scale protein backbone conformations and local structural motifs.

2. **One-shot Decoding:** Replacing the hierarchical one-shot decoding with iterative autoregressive decoding results in the most significant performance drops, particularly for single-chain proteins. This validates that one-shot generation is crucial to avoid error accumulation, especially in complex protein architectures.

3. **HFSM:** Removing HFSM leads to notable performance degradation, particularly affecting the model's ability to handle short protein sequences. Without HFSM, only a small fraction of the computational pathway is activated, as all meaningful tokens are concentrated at the beginning of the sequence. This confirms that hierarchical structural decomposition enables adaptive modeling of proteins with varying complexity and length scales, allowing the model to dynamically allocate computational resources to structurally important regions.

## 5 CONCLUSION

This work introduces FractalFold, a novel transformer-based model that leverages the inherent fractal nature of protein structures for inverse folding through recursive multi-scale decoding and hierarchical segmentation. The fractal architecture enables coarse-to-fine refinement that effectively eliminates error accumulation, while the Hierarchical Fractal Segmentation Module enables adaptive multi-scale decomposition of quasi-fractal proteins. Our approach achieves state-of-the-art performance on established benchmarks, surpassing existing methods in sequence recovery rate, perplexity, and foldability. By demonstrating that architectural alignment with biological priors can exceed brute-force data scaling, FractalFold establishes a new paradigm for structure-informed protein design and opens promising avenues for extending fractal modeling to multi-chain complexes, protein interactions, and other structural biology tasks requiring multi-scale understanding.

REPRODUCIBILITY STATEMENT

To ensure reproducibility of our results, we provide comprehensive implementation details and experimental configurations throughout this work. Section 5 contains detailed training procedures including hyperparameters, optimization settings, and hardware specifications. The complete model architecture specifications, including the fractal hierarchy design and multi-scale encoding components, are described in Section 4 with additional implementation details provided in the appendix. The codebase implementing FractalFold, including training scripts, model definitions, and evaluation pipelines, will be made publicly available upon publication.

ETHICS STATEMENT

This work adheres to the ICLR Code of Ethics. Our research uses publicly available protein structural data from the CATH dataset with no sensitive information or human subjects involved. The proposed computational method for protein sequence prediction poses no known ethical concerns and has potential benefits for drug discovery and protein engineering. We declare no conflicts of interest and ensure transparent reporting of methodology and limitations.

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

# A NOTATION

This section provides a comprehensive overview of the mathematical notation used throughout the paper, organized by the main components of our FractalFold framework.

Table 4: Mathematical Notation

| Symbol | Description |
|---|---|
| *Problem Formulation* | |
| $N$ | Protein sequence length |
| $N_{\text{atom}}$ | Number of backbone atoms per residue |
| $S \in \mathbb{R}^{N \times N_{\text{atom}} \times 3}$ | 3D coordinates of backbone atoms |
| $S_i \in \mathbb{R}^{N_{\text{atom}} \times 3}$ | Atomic coordinates of the $i$-th residue |
| $A = (a_1, a_2, \ldots, a_N)$ | Amino acid sequence |
| $a_i \in \{1, 2, \ldots, 20\}$ | Integer index for $i$-th amino acid |
| $f_\theta$ | Neural network with parameters $\theta$ |
| $\Delta^{20 \times N}$ | Space of probability distributions over sequences |
| $\mathcal{L}(\theta)$ | Conditional log-likelihood objective |
| $\mathcal{D}$ | Training dataset of structure-sequence pairs |
| *Hierarchical Modeling* | |
| $K$ | Number of fractal scales |
| $k \in \{1, 2, \ldots, K\}$ | Fractal scale index |
| $L_k$ | Number of segments at scale $k$ |
| $T^{(k)} = \{\mathbf{t}_1^{(k)}, \ldots, \mathbf{t}_{L_k}^{(k)}\}$ | Padded segments at scale $k$ |
| $\mathbf{t}_m^{(k)}$ | $m$-th segment at scale $k$ |
| $b_m, e_m$ | Start and end positions of segment $m$ |
| $H^{(k)} = \{\mathbf{h}_1^{(k)}, \ldots, \mathbf{h}_{L_k}^{(k)}\}$ | Structural features for segments at scale $k$ |
| $\mathcal{H}^{(k)}, \mathcal{T}^{(k)}$ | Hierarchical representations at scale $k$ |
| *Hierarchical Fractal Segmentation* | |
| $\mathbf{A} \in \mathbb{R}^{N \times N}$ | ESM2 attention matrix |
| $a_i$ | Link probability between residues $i$ and $i + 1$ |
| $r = L_k / L_{k-1} - 1$ | New breakpoints per segment at scale $k$ |
| $T^{(k)*}$ | Optimal segmentation at scale $k$ |
| $\langle \text{PAD} \rangle$ | Padding token |
| *Fractal Transformer* | |
| $B$ | Batch size |
| $D$ | Feature dimension |
| $\mathbf{X}^{(k)}$ | Input to transformer at scale $k$ |
| $\mathbf{s}_i^{(k)}$ | $i$-th structural token at scale $k$ |
| $\mathbf{c}^{(k-1)}$ | Conditioning token from scale $k - 1$ |
| $\mathbf{Y}^{(k)} \in \mathbb{R}^{B \times (L_k+1) \times D}$ | Transformer output at scale $k$ |
| $y_i$ | Ground truth token at position $i$ |
| $\theta$ | FractalFold model parameters |

# B LARGE LANGUAGE MODEL USAGE STATEMENT

Large Language Models were used in a limited capacity during the preparation of this manuscript. Specifically, LLMs were employed solely as auxiliary tools for grammar checking, language polishing, and minor stylistic improvements. The LLMs did not contribute to research ideation, methodology development, experimental design, data analysis, or the generation of scientific content. All research concepts, technical contributions, experimental results, and scientific insights presented in this work are entirely the product of the authors' original research and intellectual effort.

The authors take full responsibility for all content in this manuscript, including any text that may have been refined using LLM assistance for grammatical or stylistic purposes.

## C  RELATED WORKS

### C.1  PROTEIN INVERSE FOLDING

The advent of deep learning has spurred a revolution in modeling protein folding (Jumper et al., 2021; Lin et al., 2023), while the inverse problem of protein folding, which aims to infer amino acid sequences that fold into desired structures, has gained increasing attention (Dauparas et al., 2022). By representing protein backbone structures as $k$-NN graphs, geometric deep learning has achieved remarkable progress in learning inverse folding (Ingraham et al., 2019b; Dauparas et al., 2022; Hsu et al., 2022b), surpassing traditional physics-based approaches (Alford et al., 2017) and facilitating the design of experimentally validated proteins (Dauparas et al., 2022; Watson et al., 2023). Current methods primarily follow three generation strategies: autoregressive approaches like GraphTrans (Ingraham et al., 2019b), ProteinMPNN (Dauparas et al., 2022), GVP (Jing et al., 2021), and ESM-IF (Hsu et al., 2022b) that generate sequences token-by-token but suffer from slow inference speed; one-shot methods such as PiFold (Gao et al., 2023) and DE-NOVO (Mao et al., 2024) that facilitate parallel generation of multiple tokens but struggle with global consistency; and iterative refinement techniques including LM-Design (Zheng et al., 2023b), KW-Design (Gao et al., 2024), ChromaDesign (Ingraham et al., 2023), CarbonDesign (Ren et al., 2024a), and diffusion-based GraDe-IF (Yi et al., 2023b) that progressively improve predictions through multiple refinement steps, with some leveraging discrete denoising diffusion probabilistic models (Austin et al., 2021) to encompass diverse plausible solutions.Despite significant progress in computational biology, existing methods predominantly treat protein structures as flat sequential representations or simple graph topologies, failing to capture the inherent hierarchical and fractal organization of complex biological protein architectures.

### C.2  FRACTAL MODELING

Protein structures exhibit hierarchical organization from secondary motifs to tertiary architectures, resembling fractals' self-similar properties (Mandelbrot, 1983; Enright & Leitner, 2005). Current inverse folding methods treat proteins as flat sequences or simple graphs, ignoring this structural hierarchy. Hierarchical approaches have proven effective in computer vision (Burt & Adelson, 1987; Lin et al., 2017; Liu et al., 2021; Li et al., 2025) and generative modeling through cascaded diffusion (Ramesh et al., 2022; Saharia et al., 2022) and scale-space methods (Tian et al., 2024; Tang et al., 2024) that generate progressively from coarse to fine scales. To the best of our knowledge, FractalFold is the first protein inverse folding method that explicitly leverages fractal-based generation to align with the inherent fractal structure prior of proteins, modeling recursive structural hierarchy through coarse-to-fine refinement to enable biologically informed sequence design.

## D  RESULTS VISUALIZATION

This section presents detailed visualizations of FractalFold's performance on selected proteins from the CATH 4.2 test set. The results demonstrate the model's ability to generate high-quality structural predictions across diverse protein families and topologies.

Figure 4 provides a comprehensive comparison between ground truth structures and predictions from both PiFold and FractalFold methods, illustrating the superior accuracy achieved by our approach. The subsequent figures showcase individual FractalFold predictions for representative proteins, highlighting the model's consistent performance across different protein sizes and structural complexities.

Each prediction visualization includes four key components: front and side views of the predicted structure, confidence-colored representations indicating prediction reliability, and per-residue confidence plots with accompanying performance metrics. The confidence coloring scheme ranges from blue (low confidence, pLDDT < 50) to red (high confidence, pLDDT > 90), providing immediate visual feedback on prediction quality.

The selected examples span protein lengths from 114 to 195 residues and demonstrate consistently high average pLDDT scores (92.2-96.9), pTM scores (0.860-0.910), and FractalFold scores (0.664-0.685), validating the robustness of our method across the CATH 4.2 test set.

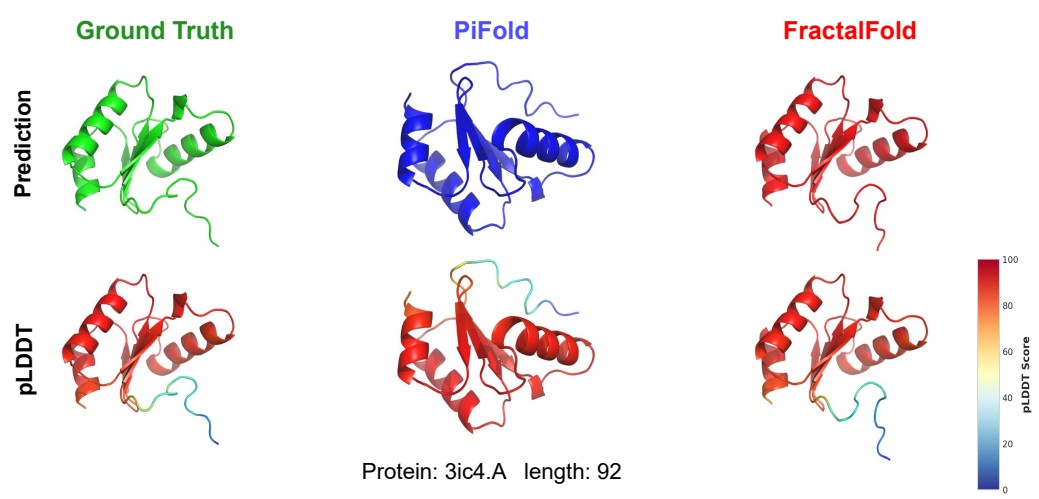

Figure 4: Comparison of protein folding predictions for 3ic4.A showing structural models (top) and confidence scores (bottom) for Ground Truth, PiFold, and FractalFold methods.

**4owz.A - FRACTAL Model**

| Metric | Value |
|---|---|
| Average pLDDT | 96.2 |
| pTM Score | 0.880 |
| FractalFold Score | 0.664 |

*Protein ID: 4owz.A | Type: FRACTAL | Length: 134*

Figure 5: FractalFold prediction for protein 4owz.A (134 residues) showing high-confidence structural model with average pLDDT of 96.2. Structure views (left) and confidence-colored representations (right) demonstrate excellent prediction quality across most residues, with performance metrics including pTM score of 0.880 and FractalFold score of 0.664.

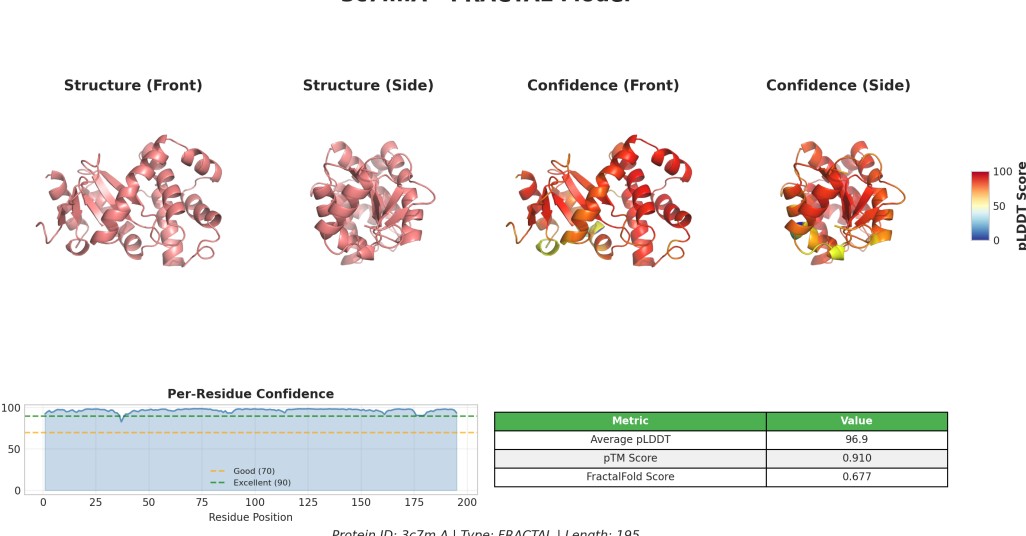

Figure 6: FractalFold prediction for protein 4lqb.A (130 residues) showing high-confidence structural model with average pLDDT of 95.0. Structure views (left) and confidence-colored representations (right) demonstrate excellent prediction quality across most residues, with performance metrics including pTM score of 0.860 and FractalFold score of 0.669.

**3c7m.A - FRACTAL Model**

Figure 7: FractalFold prediction for protein 3c7m.A (195 residues) showing high-confidence structural model with average pLDDT of 96.9. Structure views (left) and confidence-colored representations (right) demonstrate excellent prediction quality across most residues, with performance metrics including pTM score of 0.910 and FractalFold score of 0.677.

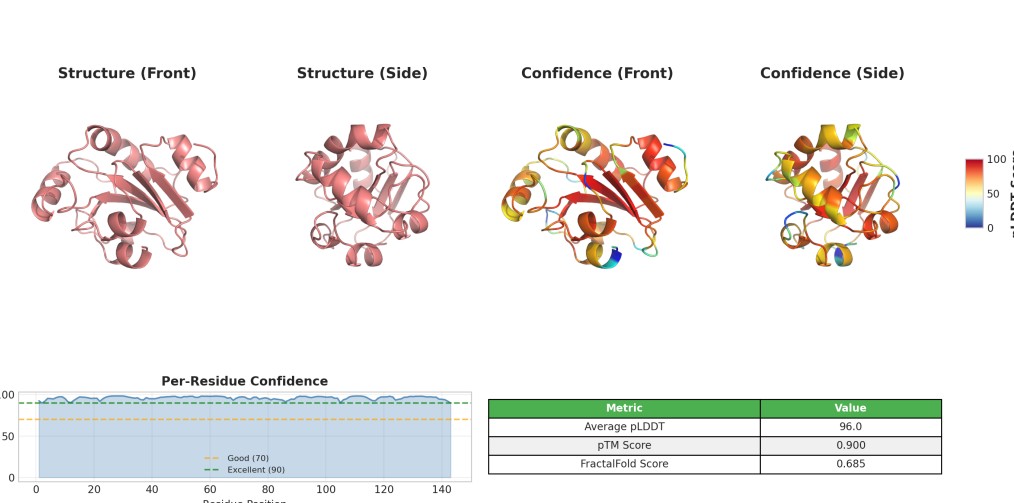

Figure 8: FractalFold prediction for protein 2fy6.A (143 residues) showing high-confidence structural model with average pLDDT of 96.0. Structure views (left) and confidence-colored representations (right) demonstrate excellent prediction quality across most residues, with performance metrics including pTM score of 0.900 and FractalFold score of 0.685.

# E   PARAMETER SENSITIVITY ANALYSIS

We conducted parameter sensitivity analysis on CATH 4.2 to examine the impact of three critical hyperparameters on FractalFold's performance. Each parameter was varied independently while maintaining baseline values for others, with results averaged across three random seeds.

## E.1   RESULTS AND ANALYSIS

Figure 9 presents the sensitivity analysis results across decode temperature, ESM-IF attention layer selection, and embedding dimension.

**Decode Temperature:** Recovery rates increase monotonically from 53.30% (temperature 10) to 53.62% (temperature 400). Higher temperatures introduce beneficial stochasticity during sequence generation, enabling broader exploration of the solution space and improved sequence-structure compatibility.

**Attention Layer Selection:** Early attention layers prove most effective, with layer 1 achieving optimal performance (53.62%) compared to deeper layers (53.41% for layer 2, 53.27% for layer 3). This indicates that early layers from pre-trained structure encoders capture the most relevant geometric information for inverse folding tasks.

**Embedding Dimension:** Moderate capacity yields optimal results, with 256 dimensions achieving peak performance (53.62%) compared to 128D (53.53%) and 384D (53.60%). This demonstrates an effective balance between representational power and generalization capability.

## E.2   IMPLICATIONS

The analysis reveals that FractalFold exhibits robust performance across parameter ranges, with the fractal architecture providing inherent stability. The findings recommend using higher decode temperatures, early attention layers from pre-trained encoders, and moderate embedding dimensions for optimal performance.

Figure 9: Parameter sensitivity analysis on CATH 4.2 dataset. Higher temperatures improve sampling diversity, early attention layers capture optimal structural information, and moderate embedding dimensions provide the best performance balance.

## F  IMPLEMENTATION DETAILS

We implemented our FractalFold model using PyTorch 2.2.2 with CUDA 11.8 runtime, leveraging mixed precision training through PyTorch's Automatic Mixed Precision (AMP) for computational efficiency. All experiments were conducted on four NVIDIA GTX 4090 GPUs

For sequence tokenization, we employed the ESM-1b alphabet with special tokens to build token IDs and masking. The model architecture follows a hierarchical design with amino acid patch sizes of 64, 8, and 1 residues, where each value represents the patch size at different hierarchical levels with 1 corresponding to single amino acid tokens. The embedding dimension was set to 256 across all levels, while the number of transformer blocks was configured as 4, 4, and 2 blocks respectively, indicating the computational depth at each hierarchical level from coarse to fine resolution.

For optimization, we employed the AdamW optimizer with $\beta_1 = 0.9$, $\beta_2 = 0.95$, and weight decay of 0.05. The learning rate was scaled based on global batch size using lr = blr × (global_batch/256) with a base learning rate of $5 \times 10^{-5}$. The learning rate schedule included linear warmup for 5 epochs followed by cosine decay to zero over the training duration.

Training utilized mixed precision with gradient clipping at a global norm of 3.0 to ensure stable convergence. Batch formation employed fixed token-budget batching with a maximum of 15,000 tokens per batch while respecting nominal batch size constraints. The complete training process was conducted for 50 epochs.

Model performance was evaluated using sequence recovery rate, computed as elementwise equality between predicted and ground truth amino acids at positions with valid coordinates:

$$\text{Recovery Rate} = \frac{1}{|\mathcal{M}|} \sum_{i \in \mathcal{M}} \mathbf{1}[\hat{y}_i = y_i] \tag{9}$$

where $\mathcal{M}$ denotes the set of positions with valid coordinates, $\hat{y}_i$ and $y_i$ are the predicted and ground truth amino acids at position $i$, respectively, and $\mathbf{1}[\cdot]$ is the indicator function which excludes padding tokens. Perplexity was also reported as an additional metric. During evaluation, the model performed non-autoregressive sampling using softmax-multinomial sampling with temperature control.

## G  PROOF OF OPTIMIZATION EQUIVALENCE

We prove that the optimization problem in Equation (X) is equivalent to selecting the $L_k - 1$ positions with the lowest link probabilities as breakpoints.

**Lemma 1** *For a sequence of length $N$ with link probabilities $\{a_i\}_{i=1}^{N-1}$, the segmentation that minimizes $\sum_{m=1}^{L_k} \sum_{i=b_m}^{e_m-1} (-\log a_i)$ is equivalent to selecting the $L_k - 1$ positions with the smallest link probabilities as breakpoints.*

**Proof G.1** *Consider the objective function:*

$$\mathcal{L}(T^{(k)}) = \sum_{m=1}^{L_k} \sum_{i=b_m}^{e_m-1} (-\log a_i) = -\sum_{m=1}^{L_k} \sum_{i=b_m}^{e_m-1} \log a_i \tag{10}$$

*Since each position $i \in \{1, 2, \ldots, N-1\}$ appears in exactly one segment, we can rewrite this as:*

$$\mathcal{L}(T^{(k)}) = -\sum_{i=1}^{N-1} \log a_i + \sum_{j \in \mathcal{B}} \log a_j \tag{11}$$

*where $\mathcal{B} = \{j_1, j_2, \ldots, j_{L_k-1}\}$ is the set of breakpoint positions, and the second term accounts for the fact that breakpoint positions are excluded from the segments (as they represent boundaries between segments).*

*The first term $-\sum_{i=1}^{N-1} \log a_i$ is constant regardless of the choice of breakpoints. Therefore, minimizing $\mathcal{L}(T^{(k)})$ is equivalent to minimizing:*

$$\sum_{j \in \mathcal{B}} \log a_j \tag{12}$$

*Since $\log$ is a monotonically increasing function, minimizing $\sum_{j \in \mathcal{B}} \log a_j$ is equivalent to minimizing $\sum_{j \in \mathcal{B}} a_j$.*

*To minimize the sum of $L_k - 1$ values from the set $\{a_1, a_2, \ldots, a_{N-1}\}$, we must select the $L_k - 1$ smallest values. Therefore, the optimal breakpoints correspond to the positions with the $L_k - 1$ lowest link probabilities.*

This equivalence justifies our greedy approach in Algorithm 1, where we iteratively select positions with minimum link probabilities as breakpoints at each hierarchical scale.

## H  COMPUTATIONAL COMPLEXITY ANALYSIS

The computational complexity of transformer-based protein language models is fundamentally constrained by the quadratic scaling of self-attention mechanisms with respect to sequence length. For a protein sequence of length $N$ with hidden dimension $d$, the standard transformer attention mechanism requires $\mathcal{O}(N^2 d)$ operations per layer. This quadratic complexity poses significant computational barriers for long protein sequences, which can exceed 1000-5000 residues in many biologically relevant cases. The FractalFold architecture addresses this limitation through hierarchical fractal segmentation, achieving theoretical complexity reduction to $\mathcal{O}(K \cdot N \cdot \log N \cdot d)$, where $K$ represents the number of fractal scales.

For a transformer layer processing a protein sequence of length $N$, the total computational complexity per attention head is:

$$\mathcal{C}_{\text{standard}} = 2N^2 d + N^2 = N^2(2d + 1) = \mathcal{O}(N^2 d)$$

The memory complexity for storing attention matrices scales as $\mathcal{O}(N^2)$, which becomes prohibitive for long sequences.

The FractalFold architecture decomposes the global attention computation into hierarchical local attention operations across $K$ fractal scales. At scale $k$, the sequence is partitioned into $L_k$ segments, where $L_k = \min(2^k, N)$ follows exponential growth constrained by sequence length.

For each scale $k$, the average segment length is:

$$\ell_k = \lfloor N/L_k \rfloor$$

The computational complexity at scale $k$ is:

$$\mathcal{C}_k = L_k \cdot \ell_k^2 \cdot d = L_k \cdot \left(\frac{N}{L_k}\right)^2 \cdot d = \frac{N^2 d}{L_k}$$

The total complexity across all scales becomes:

$$\mathcal{C}_{\text{fractal}} = \sum_{k=1}^{K} \mathcal{C}_k = N^2 d \sum_{k=1}^{K} \frac{1}{L_k}$$

Since $L_k = \min(2^k, N)$, for sequences where $N > 2^K$, this sum approximates:

$$\sum_{k=1}^{K} \frac{1}{2^k} = 1 - \frac{1}{2^K} < 1$$

Therefore: $\mathcal{C}_{\text{fractal}} < N^2 d$, achieving complexity reduction.

For the asymptotic case where $K = \mathcal{O}(\log N)$, the complexity becomes:

$$\mathcal{C}_{\text{fractal}} = \mathcal{O}(K \cdot N \cdot \log N \cdot d)$$

The theoretical complexity reduction from $\mathcal{O}(N^2)$ to $\mathcal{O}(N \log N)$ represents a fundamental advancement in transformer scalability for protein sequence modeling, enabling the processing of previously intractable long-range protein interactions while maintaining the representational power of self-attention mechanisms.

