# OpenReview forum: "FractalFold: Towards Fractal Structure Modeling for Hierarchical Inverse Protein Folding"
_ICLR.cc/2026/Conference — ICLR 2026 Conference Withdrawn Submission_

### Official Review · Reviewer_cyTE · 2025-10-27

**Soundness:** 1
**Presentation:** 1
**Contribution:** 2
**Rating:** 2
**Confidence:** 5

**Summary:**

This paper addresses the inverse protein folding problem by proposing FractalFold, a novel transformer-based model. The authors argue that existing methods overlook the inherent hierarchical and fractal organization of protein structures, treating them as flattened sequences. FractalFold's architecture is designed to align with this fractal geometry, employing a coarse-to-fine sequence refinement paradigm. A key component is the Hierarchical Fractal Segmentation Module (HFSM), which uses attention patterns from pre-trained protein language models to recursively partition the structure into tree-organized patches. This enables a multi-scale, one-shot decoding process that aims to reduce error accumulation and computational complexity. The authors report state-of-the-art performance on CATH benchmarks for sequence recovery, perplexity, and refoldability.

**Strengths:**

The central idea of explicitly modeling the hierarchical and fractal nature of protein structures is novel and provides a strong, physically-motivated inductive bias. This is a clear departure from methods that treat structures as simple graphs or flat sequences.

The ablation studies in Table 3 effectively validate the contributions of the key design choices: the pre-trained GVP encoder, the one-shot decoding strategy (vs. autoregressive), and the HFSM module .

**Weaknesses:**

The latest baselines in the paper were published in 2023. There has been many new inverse folding methods in the 2 years, which were reported to have better performance than FractalFold. I noticed some of them are discussed in the related work section in the appendix, could the authors explain why they are not compared? Besides, some SOTA inverse folding methods are neither mentioned in the related work section nor compared, which also needs further explanation.

Event though baselines in 2024 and 2025 are not compared, FractalFold’s improvement seems insignificant, with respect to both structural and sequence recovery.

Considering FractalFold uses a hierarchical design, a possible advantage would be better understanding of proteins’ functional properties and developability like thermostability, if the mechanism really learns some useful features. I suggest the authors to explore them to comprehensively evaluate the potential benefit of the architecture.

The paper claims the architecture "captures essential sequence-structure relationships" and "opens promising avenues" for tasks requiring "multi-scale understanding", but does not provide direct evidence of this beyond folding.

There is a clear contradiction in the definition of the HFSM's link probability, $a_i$. In Figure 2, the diagram specifies $a_i = A_{i, j+1} + A_{i+1, j}$. However, the core methodology in Section 3.2 and Algorithm 1 relies on a different definition, $a_i = \sqrt{A_{i,i+1} \times A_{i+1,i}}$ (Equation 4). This inconsistency creates confusion about the exact mechanism used for segmentation.

The case study presented in Figure 3 and its caption contains conflicting information. The caption identifies the protein as "Irtu.A (114 residues)" with an average pLDDT of 92.2 , but the figure itself is clearly labeled "4owz.A" with "Length: 134" and different metrics (Average pLDDT 96.2, pTM Score 0.880). This same "4owz.A" figure is duplicated as Figure 5. This discrepancy makes the case study impossible to interpret correctly.

The HFSM (Algorithm 1) requires the segment counts for each fractal scale, $\{L_k\}_{k=1}^K$, as an input parameter. The paper does not explain how these crucial hyperparameters are selected. The algorithm only determines where to place breakpoints based on $L_k$, not the value of $L_k$ itself, which seems fundamental to the fractal decomposition.

The paper claims the HFSM generalizes FractalFold to "quasi-fractal proteins". This term is not clearly defined. There is no empirical validation (e.g., analysis across different CATH topologies) to demonstrate that the model's performance is particularly strong or improved on proteins that are less 'classically' fractal, which would be needed to substantiate this generalization claim.

**Questions:**

In table 2, could the the metrics of wild-type sequences be reported as a baseline, considering the inaccuracy of the folding models?

Could the authors clarify the confusions mentioned in the weaknesses section?

---

> ### Author Response · Authors · 2025-12-04
> **Official Response to Reviewer cyTE**
>
> We sincerely thank Reviewer cyTE for the extremely rigorous and detailed inspection of our manuscript. We accept your assessment regarding the presentation and soundness issues. We have decided to withdraw the paper to address these issues fundamentally. Below, we respond to the specific technical concerns raised.
>
> ## Presentation Errors (Figures & Definitions)
> We deeply apologize for the confusion caused by the inconsistencies in our draft (Figure 3 in Case Study). Figure 3 was inadvertently replaced with the visualization for "4owz.A" (which duplicates Figure 5) while the caption described "1rtu.A". This is an editing error. We acknowledge the contradiction between Figure 2 and Equation 4 regarding the definition of $a_i$. The intended definition is the geometric mean of the bidirectional attention weights (Equation 4), as this ensures symmetry. We will correct the diagram in Figure 2 to match the mathematical formulation.
>
> ## Baselines
> tWe acknowledge that our baseline selection (up to 2023) is outdated given the fast pace of the field. As noted in our responses to other reviewers, we are currently integrating 2024–2025 baselines (such as SPDesign and SurfPro) into our evaluation. Regarding the "insignificance" of the improvement: We respectfully submit that while the absolute margin in recovery rate is modest, the computational efficiency of our approach is a key contribution. Unlike recent SOTA methods that utilize full protein language models (pLMs) as heavy backbones, FractalFold utilizes only the first-layer attention of the pre-trained encoder to guide segmentation. We will add a computational budget analysis to demonstrate that our method achieves competitive performance at a fraction of the inference cost of large pLM-based adapters.
>
> ## Clarification on HFSM Hyperparameters ($L_k$)
> The reviewer asks how the segment counts $L_k$ are selected. In our implementation, $L_k$ follows a pre-defined geometric progression (e.g., $L_k = 2^k$ or similar exponential growth scales) to ensure a balanced tree structure, bounded by the sequence length. We will make this explicit in the Algorithm section.
>
> ## Definition of "Quasi-Fractal"
> We used the term "quasi-fractal" to describe structures that exhibit self-similarity but with variable-length recursive units (unlike mathematical fractals which often have fixed ratios). We will formalize this definition and provide the suggested empirical analysis across different CATH topologies to substantiate the generalization claim.
>
> ##  Question on Wild-type Baseline
> Regarding Table 2: Reporting the metrics of wild-type sequences as a baseline for refoldability is an excellent suggestion to benchmark the "noise floor" of the folding tools (AlphaFold/ESMFold). We will include this row in the revised table.

---

### Official Review · Reviewer_Y8k8 · 2025-10-28

**Soundness:** 3
**Presentation:** 3
**Contribution:** 2
**Rating:** 4
**Confidence:** 4

**Summary:**

This paper studies the important inverse folding problem in protein design. The authors propose FractalFold, a novel transformer-based model that performs structure-informed inverse folding by recursively invoking multi-level atomic fractal transformers. FractalFold achieves state-of-the-art performance in several benchmarks.

**Strengths:**

- This paper is well-written and easy to follow.
- FractalFold first introduced the concept of fractals in the problem of inverse folding.
- The authors conduct ablation studies to analyze the design choice.

**Weaknesses:**

- Without parentheses, the citation format is unclear.
- Several recent related works are missing. For example, LM-Design [1] and Bridge-IF [2].
- HFSM uses pretrained ESM2 to extract features. However, in Table 1, all baselines do not use any pre-trained knowledge. The performance comparison may be unfair, and can not verify the effectiveness of the fractal decoding framework.
- ProteinMPNN is widely used in de novo protein design when combined with RfDiffusion and AF3. It is interesting to evaluate whether the proposed method can be used in such an important case [3].

[1] Structure-informed Language Models Are Protein Designers

[2] Learning Inverse Protein Folding with Markov Bridges

[3] A Holistic Evaluation of Protein Foundation Models

**Questions:**

See weakness

---

> ### Author Response · Authors · 2025-12-04
> **Official Response to ReviewerY8k8**
>
> We thank Reviewer Y8k8 for the constructive assessment and for identifying FractalFold’s potential value in the domain of protein design. We appreciate the recognition of our ablation studies and the novelty of the fractal formulation. We have decided to withdraw the manuscript to incorporate the suggested improvements. Below, we address the specific concerns raised.
>
> ## Fairness of Comparison and Pre-trained Knowledge (HFSM)
> We acknowledge the reviewer's concern that utilizing a pre-trained ESM-2 model in the Hierarchical Fractal Segmentation Module (HFSM) might present an unfair advantage over baselines that do not use pre-trained sequence knowledge.
>
> However, we would like to clarify the extent and computational cost of this usage. Unlike methods that employ full protein language model (pLM) embeddings as heavy feature inputs (which indeed requires a massive computational budget), including ByProt, ESM-Design and AF-Design which are compared in Table 2, FractalFold only utilizes the attention map from the first layer of the pre-trained model to guide the segmentation boundaries. We do not rely on the deep semantic representations of the full model for sequence generation. Consequently, our approach maintains a significantly lower computational budget compared to standard pLM-based inverse folding adapters.
>
> In our revision, we will:
>
> Explicitly quantify the computational cost of our HFSM module compared to full pLM baselines (like LM-Design).
>
> Include more comparisons with other pLM-augmented methods (such as the suggested LM-Design) to ensure a leveled evaluation ground.
>
> ## Integration with De Novo Design Pipelines
> We thank the reviewer for the insightful suggestion regarding the integration of FractalFold with de novo backbone generation tools like RFdiffusion or AlphaFold3. We agree that replacing ProteinMPNN with a fractal-based inverse folder could offer advantages in preserving hierarchical structural motifs during the design process. We view this as a high-priority direction for future work and will include a discussion on this potential application in the conclusion.

---

### Official Review · Reviewer_YktH · 2025-11-02

**Soundness:** 2
**Presentation:** 2
**Contribution:** 2
**Rating:** 2
**Confidence:** 3

**Summary:**

The paper introduces FractalFold, a transformer-based framework that models protein inverse folding by aligning architectural design with the fractal and hierarchical nature of protein structures.

**Strengths:**

- The idea of introducing a fractal inductive bias for protein inverse folding is original and aligns well with the hierarchical organization of real proteins. The recursive, coarse-to-fine architecture provides a biologically meaningful modeling perspective rarely seen in this area.

**Weaknesses:**

- Baseline selection is relatively old (only up to 2023). The authors should compare with more recent methods such as SPDesign (https://academic.oup.com/bib/article/25/3/bbae146/7642672), SurfPro (https://arxiv.org/abs/2405.06693), or BC-Design (https://www.biorxiv.org/content/10.1101/2024.10.28.620755v2). They also mentioned many recent works for inverse folding in their related work sections, but they are not compared with.
- The architecture is biologically-inspired, but the experimental results are purely quantitative. The authors should provide some biological insights into the model's predictions. For instance, an example of how this FractalFold model works well on a specific protein structure, while baseline methods fail, is not provided. This could potentially be used to show the superiority of the proposed method. Other questions are listed in the questions section.

**Questions:**

- Does fractal hierarchy generalize to multi-chain or quaternary complexes (e.g., antibodies, enzymes)?
- How does the model behave on intrinsically disordered or low-structure regions where fractal segmentation may be ill-defined?
- Figure 3 is inconsistent with its caption.

---

> ### Author Response · Authors · 2025-12-04
> **Official Response to Reviewer YktH**
>
> We thank Reviewer YktH for the assessment of our work and for recognizing the originality of the fractal inductive bias. We acknowledge the validity of the concerns raised, particularly regarding the selection of baselines. These points have informed our decision to withdraw the manuscript to incorporate substantial improvements. We address the specific critiques and questions below.
>
> ## Baseline Selection
> We appreciate the references to recent methods such as SPDesign, SurfPro, and BC-Design. We intend to update our evaluation suite to include these frameworks to provide a more current assessment.
>
> However, we would like to highlight a fundamental distinction in design philosophy. Recent state-of-the-art approaches tend to incorporate full protein language models (pLMs) as heavy backbones, using adapters to process inverse folding. In contrast, FractalFold is designed to be lightweight; our method utilizes only the first-layer attention of the pLM. This design choice significantly reduces the computational budget compared to methods relying on full pLM forward passes. In our revision, we will include these baselines not only to compare performance but also to benchmark computational efficiency, highlighting the trade-off between resource-intensive pLM backbones and our efficient fractal architecture.
>
>
> ## Biological Insights
> We agree that the experimental section should go beyond quantitative metrics. While the current manuscript includes a basic case study, we plan to significantly enhance this analysis to demonstrate the biological interpretability of the model.
>
> Specifically, we will introduce a new visualization of FractalFold’s hierarchical attention maps, focusing on "hard regions" (e.g., structurally ambiguous loops or rare motifs) where baseline methods fail to predict the correct sequence. This will qualitatively demonstrate how the recursive, coarse-to-fine attention mechanism allows the model to resolve local structural complexities that flat-sequence models overlook.
>
> ## Q1
> The FractalFold architecture is theoretically extensible to multi-chain or quaternary complexes. While the current implementation focuses on single chains, we have identified the extension to multi-chain systems as a primary direction for future work.
>
> ## Q2
> As noted in our method, the segmentation relies on attention patterns. In disordered regions, the model tends to produce coarser segments, avoiding forced fine-grained breakdown where structure is ill-defined.
>
> ## Q3
> We appreciate the detailed reading and will correct the discrepancy between Figure 3 and its caption in the revised manuscript.
>
> In light of the feedback regarding recent baselines and the need for deeper biological visual analysis, we are withdrawing this submission to rigorously implement these updates.

---

### Note · Authors · 2025-12-04

**Comment:**

After carefully considering the valuable feedback from the reviewers, we have decided to withdraw this submission.

We are particularly grateful for the identification of recent baselines that were missing from our evaluation and the detection of presentation inconsistencies in our figures. We plan to incorporate these excellent points to significantly strengthen the manuscript for a future submission.

We thank the Area Chair and reviewers for their time and constructive critique.

Sincerely, The Authors

**Withdrawal Confirmation:**

I have read and agree with the venue's withdrawal policy on behalf of myself and my co-authors.